# Host Metabolic Changes during Mycobacterium Tuberculosis Infection Cause Insulin Resistance in Adult Mice

**DOI:** 10.3390/jcm11061646

**Published:** 2022-03-16

**Authors:** Neelam Oswal, Kezia Lizardo, Dhanya Dhanyalayam, Janeesh P. Ayyappan, Hariprasad Thangavel, Scott K. Heysell, Jyothi F. Nagajyothi

**Affiliations:** 1Center for Discovery and Innovation, Hackensack University Medical Center, Hackensack, NJ 07110, USA; neelam.oswal@hmh-cdi.org (N.O.); kezia.lizardo@hmh-cdi.org (K.L.); dhanya.dhanyalayam@hmh-cdi.org (D.D.); hariprasad.thangavel@hmh-cdi.org (H.T.); 2Department of Biochemisty, University of Kerala, Thiruvananthapuram 695034, Kerala, India; janeeshbiochemistry@keralauniversity.ac.in; 3Division of Infectious Diseases and International Health, University of Virginia, Charlottesville, VA 22908, USA; skh8r@hscmail.mcc.virginia.edu

**Keywords:** tuberculosis, type 2 diabetes, diet and age, insulin resistance, adipose tissue, inflammation, lipolysis, lipid and energy metabolism

## Abstract

Tuberculosis (TB) is a highly infectious bacterial disease that primarily attacks the lungs. TB is manifested either as latent TB infection (LTBI) or active TB disease, the latter posing a greater threat to life. The risk of developing active TB disease from LTBI is three times higher in individuals with type 2 diabetes mellitus (T2DM). The association between TB and T2DM is becoming more prominent as T2DM is rapidly increasing in settings where TB is endemic. T2DM is a chronic metabolic disorder characterized by elevated blood glucose, insulin resistance, and relative insulin deficiency. Insulin resistance and stress-induced hyperglycemia have been shown to be increased by TB and to return to normal upon treatment. Previously, we demonstrated that adipocytes (or fat tissue) regulate pulmonary pathology, inflammation, and *Mycobacterium tuberculosis* (*Mtb*) load in a murine model of TB. Metabolic disturbances of adipose tissue and/or adipocyte dysfunction contribute to the pathogenesis of T2DM. Thus, pathological adipocytes not only regulate pulmonary pathology, but also increase the risk for T2DM during TB infection. However, the cellular and molecular mechanisms driving the interaction between hyperglycemia, T2DM and TB remain poorly understood. Here, we report the impact of *Mtb* infection on the development of insulin resistance in mice fed on a regular diet (RD) versus high-fat diet (HFD) and, conversely, the effect of hyperglycemia on pulmonary pathogenesis in juvenile and adult mouse models. Overall, our study demonstrated that *Mtb* persists in adipose tissue and that *Mtb* infection induces irregular adipocyte lipolysis and loss of fat cells via different pathways in RD- and HFD-fed mice. In RD-fed mice, the levels of TNFα and HSL (hormone sensitive lipase) play an important role whereas in HFD-fed mice, ATGL (adipose triglyceride lipase) plays a major role in regulating adipocyte lipolysis and apoptosis during *Mtb* infection in adult mice. We also showed that *Mtb* infected adult mice that were fed an RD developed insulin resistance similar to infected adult mice that were overweight due to a HFD diet. Importantly, we found that a consequence of *Mtb* infection was increased lipid accumulation in the lungs, which altered cellular energy metabolism by inhibiting major energy signaling pathways such as insulin, AMPK and mToR. Thus, an altered balance between lipid metabolism and glucose metabolism in adipose tissue and other organs including the lungs may be an important component of the link between *Mtb* infection and subsequent metabolic syndrome.

## 1. Introduction

In non-pandemic periods, tuberculosis (TB) caused by the bacterium *Mycobacterium tuberculosis* (*Mtb*), represents the leading cause of death from a single infectious disease worldwide. According to the World Health Organization (WHO), 10 million people contracted TB and 1.4 million died from the disease in 2019. A further one-fourth of the world’s population is estimated to be infected with *Mtb*, where the pathogen is living latently in the host but not causing symptomatic disease [1]. The risk of developing active disease, termed reactivation, in adults with latent TB infection (LTBI) is greater in people with type 2 diabetes mellitus (T2DM). T2DM and TB epidemics are converging in co-endemic regions of high population density such as China and India [2,3]. Currently 10% of global TB cases are linked to T2DM, and by 2030 this proportion is expected to surge [2,4].

Multiple meta-analyses of clinical research data and epidemiological studies have shown that T2DM increases the risk of TB reactivation [5,6,7,8], and also that those with diabetes and active TB have more severe manifestations of pulmonary TB with increased incidence of cavitary lesions [9], take a longer time for microbiological response to TB treatment [10], and suffer a higher mortality compared to TB patients without diabetes [11,12]. An inadequate response to TB treatment also increases the risk of acquiring *Mtb* drug resistance. Furthermore, in some limited studies the risk of extrapulmonary TB dissemination was higher in patients with T2DM compared to those without T2DM [13].

T2DM is a chronic metabolic disorder that affects the functions of pancreatic beta cells leading to insulin deficiency. While chronic T2DM is associated with insulin deficiency related increase in blood glucose, the initial transient period of T2DM is associated with insulin resistance. Despite the clinical evidence for a pathological interaction between TB and T2DM, the cellular and molecular mechanisms driving these interactions remain poorly understood [14,15]. T2DM usually develops in adults over the age of 45 years. Although TB is a risk for any age group, most infected individuals diagnosed with TB in developed nations, including the United States and Europe, are ≥65 years and ≥50 years of age, respectively [16,17]. Similar increases in the incidence of TB have been demonstrated in association with advancing age (≥45 years) in TB endemic regions, such as China and India [18,19]. Thus, TB affects adults in their more productive years [20]. Age is also an important risk factor for T2DM, and T2DM-related TB occurs at older ages relative to non-T2DM related TB (approximately ≥40 years of age) [21]. Recent epidemiological studies also indicate that TB disease may influence the pathogenesis and incidence of diabetes in *Mtb* infected patients due to stress and/or increased proinflammatory signaling [22]. Therefore, further studies in animal TB models are necessary to understand the mechanistic link between T2DM, TB and insulin resistance and to correlate the clinical data.

We previously demonstrated that adipocytes/adipose tissue regulate pulmonary pathology, inflammation, and bacterial load in a murine TB model [23]. Furthermore, deregulated adipocyte/adipose tissue physiology has been shown to be involved in the pathogenesis of metabolic disorder and T2DM [24,25]. Therefore, pathological adipose tissue may play a major role not only in regulating pulmonary pathology, but also in increasing the risk of developing metabolic disorder during infection and increasing the risk for subsequent T2DM. Since the causal and mechanistic interactions between metabolic disorder and *Mtb* infection are not well understood, we investigated the impact of *Mtb* infection on the development of insulin resistance in mice fed either a regular diet (RD) or high-fat diet (HFD) and the effect of diet-induced hyperglycemia on pulmonary pathogenesis. The current study examines: (i) the effect of diet on pulmonary pathology; (ii) the effect of *Mtb* infection on the development of insulin resistance in juvenile and adult mice, and (iii) the role of adipose tissue in regulating pulmonary pathology and insulin sensitivity during *Mtb* infection in mice of different age groups and fed either a control carbohydrate rich diet or a high fat diet.

## 2. Materials and Methods

Animal ethics statement: All animal experimental protocols were approved by the institutional Animal Care and Use and Institutional Biosafety Committees of Rutgers University and Center for Discovery and Innovation of Hackensack University Medical Center and adhere to National Research Council guidelines.

Animal model and experimental design: The C57BL/6 (4 weeks old, male, *n* = 30) mice were purchased from the Jackson Laboratory and housed in the BSL2 facility at the Public Health Research Institute Center, New Jersey Medical School, Rutgers University. Mice were maintained on a 12-h light/12-h dark cycle and housed in groups of two to four mice per cage. Mice were divided into two groups at 4-weeks age (*n* = 15 per group) and fed on either a high-fat diet (HFD; 60% fat calories (20% calories of protein and 20% calories of carbohydrates) D12492 Research Diets, Inc., New Brunswick, NJ, USA) or a low-fat control diet (RD; 10% calories of fat). Although RD diet was designed to be used as a control diet, it is also considered a carbohydrate-rich diet (RD; 70% carbohydrate calories (20% calories of protein and 10% calories of fat) when compared to the “standard” rodent diet PicoLab #5058. When these mice were at 8 weeks old, we purchased another batch of C57Bl/6 mice (4 weeks old, male, *n* = 30) and divided them (second batch) into two groups (*n* = 15/group) and fed on either a HFD or RD for 2 weeks. The mice, 10 weeks age (the first batch mice) were considered as adult group and 6 weeks (the second batch mice) as juvenile group. At 6-weeks age (juvenile mice) and 10-weeks age (adult mice), half of the RD and HFD mice were aerosol infected with M. tuberculosis H37Rv as described earlier [23]. Briefly, M. tuberculosis aerosols were generated by a Lovelace nebulizer (In-tox Products, Albuquerque, NM, USA) with a 10-mL bacterial suspension of about 1 × 10^6^ bacilli/mL in saline containing 0.04% Tween 80, and the mice were exposed to the aerosol for 30 min, which results in seeding of approximately 100 colonizing CFU per lung. Uninfected mice were fed on either HFD (*n* = 7/batch) or RD (*n* = 7/batch) and used as respective controls in all the experiments. Mice were weighed once every week. Serum samples were obtained from 75 μL of blood collected from the orbital venous sinus (using isoflurane anesthesia) 30 DPI and mice were sacrificed and lungs and visceral fat pads (epididymal fat pads) were harvested. The weights of harvested visceral fat pads were noted. Portions of the tissues were homogenized in phosphate-buffered saline-Tween (PBST), and serial dilutions of the homogenates were plated onto Middlebrook 7H10 agar (Difco BD, Sparks, MD, USA) to determine the number of bacterial CFU. Portions of the harvested tissues were fixed in 10% formalin for histological analysis. Portions of tissues were also stored immediately at 80 °C for protein extraction. The experiment was repeated using the same number of mice. A flowchart describing the experimental design is presented as Appendix A.

Glucose, Insulin and HOMA-IR (Homeostatic Model Assessment for Insulin Resistance) measurements: Mice were fasted for 12 h and blood was drawn from the tail vein at 30 dpi and blood glucose levels were measured using a OneTouch glucometer. Plasma insulin concentrations were measured in terminal blood samples using a Mouse Insulin ELISA kit (Ultra Sensitive, Crystal Chem). The homeostasis model for insulin resistance (HOMA-IR) was calculated from the fasting blood glucose (mmol/L) × fasting plasma insulin (µU/mL) divided by 22.5 [26].

Immunoblot analysis: Protein lysates of the lung and adipose tissue samples were prepared and immunoblot analysis performed as described earlier [23]. Insulin Receptor β-specific rabbit monoclonal antibody (#4B8, 1:1000 dilution, Cell Signaling Technology, Boston, MA, USA), Phospho-IRS1 (S307)-specific rabbit polyclonal antibody (#ab5599, 1:1000 dilution, Abcam, Boston, MA, USA), Phospho-Akt (Ser473)-specific rabbit monoclonal antibody (#D9E, 1:1000 dilution, Cell Signaling), Phospho-GSK3β (Ser9) rabbit monoclonal antibody (#D85E12, 1:1000 dilution, Cell Signaling), Phospho-mTOR (Ser2448) rabbit monoclonal antibody (#D9C2, 1:1000 dilution, Cell Signaling), mTOR rabbit monoclonal antibody (#7C10, 1:1000 dilution, Cell Signaling), AMPKα rabbit polyclonal antibody (1:1000 dilution, Cell Signaling), Phospho-AMPKα (Thr172) rabbit polyclonal antibody (#AB_2554429, 1:1000 dilution, Thermo Fisher Scientific, Waltham, MA, USA), Hexokinase II-specific monoclonal antibody (#C64G5, 1:1000 dilution, Cell Signaling), AceCS1-specific rabbit monoclonal antibody (#D19C6, 1:1000 dilution, Cell Signaling), Phospho-Acetyl-CoA Carboxylase (Ser79) specific rabbit monoclonal antibody (#D7D11, 1:1000 dilution, Cell Signaling), Acetyl-CoA Carboxylase specific rabbit monoclonal antibody (#C83B10, 1:1000 dilution, Cell Signaling), Pyruvate dehydrogenase specific rabbit polyclonal antibody (#2784, 1:1000 dilution, Cell Signaling), Fatty acid synthase-specific rabbit monoclonal antibody (#C20G5, 1:1000 dilution, Cell Signaling), Phospho-ATP-Citrate Lyase (Ser455)- specific rabbit polyclonal antibody (#4331, 1:1000 dilution, Cell Signaling), ATP-Citrate Lyase—specific rabbit polyclonal antibody (#4332, 1:1000 dilution, Cell Signaling), SREBP1-specific rabbit polyclonal antibody (#ab28481, 1:1000 dilution, Abcam), PPARα-specific mouse monoclonal antibody (#MA1-822, 1:1000 dilution; Thermo Fisher Scientific, Waltham, MA, USA), phospho-HSL(Ser563)-specific rabbit monoclonal antibody (#4139, 1:1000 dilution; Cell Signaling), ATGL-specific rabbit monoclonal antibody (#30A4, 1:1000 dilution; Cell Signaling), Perilipin 1-specific rabbit monoclonal antibody (#D1D8, 1:1000 dilution; Cell Signaling), Phospho-Perilipin 1 (Ser522)-specific mouse monoclonal antibody (#4856, 1:2500 dilution; Vala Sciences, San Diego, CA, USA), ACSL1- specific rabbit polyclonal antibody (#4047, 1:1000 dilution; Cell Signaling), CPT1A-specific mouse monoclonal antibody (#8F6AE9, 1:2000 dilution, Abcam), F4/80-specific rat monoclonal (#CI-A3-1, 1:1000 dilution, Novus Biologicals, Littleton, CO, USA), PPARγ-specific rabbit monoclonal antibody (#C26H12, 1:1000 dilution; Cell Signaling), Adiponectin-specific mouse monoclonal antibody (#19F1, 1:1000 dilution; Abcam), IFNγ-specific rabbit monoclonal antibody (#EPR1108, 1:1000 dilution; Abcam), TNFα-specific rabbit polyclonal antibody (1:1000 dilution, Abcam).

Histological analyses: Freshly isolated tissues were fixed with phosphate-buffered formalin for a minimum of 48 h and then embedded in paraffin wax. H&E staining was performed, and the images were captured as previously published [23]. Four to six images per section of each lung or WAT were scored blindly. For each sample, histologic evidence of pathology was classified in terms of the presence of infiltrated immune cells, lipid droplets, and foamy macrophages and was graded on a 6-point scale ranging from 0 to 5. Auramine-rhodamine staining of the lung and adipose tissue sections was performed, and the images were captured [23]. Four to six images per section of each lung were quantitated by manually counting the stained bacteria.

Elisa analyses: Serum levels of Insulin (Invitrogen, Waltham, MA, USA), Triglyceride (Fisher Scientific), Fatty Acid (Abcam), HDL, LDL/VLDL and total cholesterol (Avantor, Radnor, PA, USA) were measured using ELISA kits according to the manufacturer’s protocol.

Statistical data analysis: Statistical analysis was performed using GraphPad Prism (GraphPad Software, Inc., La Jolla, CA, USA). Uninfected RD fed mice served as control for data analysis. Comparisons between groups were made using unpaired Student’s t-test and One-Way ANOVA as appropriate. Values of *p* < 0.05 were considered statistically significant.

## 3. Results

*Mtb* infection causes weight loss in adult but not juvenile mice: To analyze the effect of HFD on TB pathogenesis in juvenile and adult mice, we fed either HFD or RD to mice starting at 4 weeks of age and then infected one set of mice at 6 weeks of age (juvenile) and the other set at 10 weeks of age (adult). Mice fed a HFD showed increased (*p* ≤ 0.01) body weight compared to RD fed mice at 6 weeks of age (19.8 ± 1.8 g vs. 17.5 ± 1.1 g) and 10 weeks of age (28 ± 1.6 g vs. 23 ± 2 g) (data not shown). RD and HFD fed uninfected adult mice showed significantly increased body weight (5.5 ± 0.9 g and 8.2 ± 0.2, respectively; *p* ≤ 0.01) and expanded visceral adipose tissue (1.5-fold and 2.5-fold, respectively; *p* ≤ 0.01) compared to their respective diet fed uninfected juvenile mice (data not shown). At 4 weeks post infection (wpi), *Mtb* infected juvenile mice showed no significant differences in their body weights compared to their respective diet fed uninfected mice (RD 23.2 ± 2.0 g and HFD 27.6 ± 1.7 g). However, the infected adult mice showed a significant decrease (*p* ≤ 0.05) in their body weights compared to their respective diet fed uninfected groups (uninfected RD vs. infected RD (26.2 ± 1.7 g vs. 22.5 ± 1.8 g) and uninfected HFD vs. infected HFD (33.7 ± 2.2 g vs. 28.9 ± 1.9 g) at 4 wpi.

Development of insulin resistance during *Mtb* infection in mice is age dependent: We analyzed the effects of different diets on the basal glucose and insulin levels 4 weeks post infection (wpi) in mice (juvenile and adult mice infected at 6 and 10 weeks of age, respectively). As expected, HFD fed uninfected mice showed significantly increased basal glucose (but not insulin) levels compared to uninfected RD fed mice (control group) in both juvenile and adult mice (*p* < 0.05 and *p* < 0.01, respectively; Figure 1a,b). In juvenile mice, the levels of basal glucose and insulin were not altered in *Mtb* infected mice at 4 wpi compared to control mice (Figure 1a). The HOMA IR (Homeostatic Model Assessment for Insulin Resistance) calculated based on the basal glucose and insulin levels showed no significant difference in both infected and uninfected HFD mice compared to control mice (juvenile control) (Figure 1a). In adult mice, the levels of basal glucose were altered at 4 wpi; basal glucose levels were significantly higher in infected mice irrespective of the diets (infected RD fed mice (*p* ≤ 0.05) and infected HFD mice (*p* ≤ 0.001)) compared to control mice, and basal insulin levels were significantly higher in *Mtb* infected HFD fed mice (*p* ≤ 0.01) compared to control mice (Figure 1b). The HOMA-IR calculated based on the basal glucose and insulin levels was significantly higher in infected mice irrespective of the diets fed and significantly higher (*p* ≤ 0.01) in infected HFD fed mice compared to infected RD fed mice (Figure 1b). Between the HFD-fed groups, the levels of basal glucose and insulin were not altered in *Mtb* infected HFD fed juvenile mice compared to uninfected HFD fed juvenile mice. However, in adult mice, the levels of insulin and HOMA-IR levels significantly increased (*p* ≤ 0.05) in infected HFD-fed mice compared to uninfected HFD-fed mice (Figure 1b). These data suggest that *Mtb* infection induces systemic insulin resistance in adult mice and that HFD further increases insulin resistance during *Mtb* infection.

Diet regulates *Mtb* load and pathology in the lungs and adipose tissue differently in juvenile and adult mice: We analyzed the effect of HFD on lung bacterial burden in aerosol *Mtb* infected (H37Rv 10^6^ bacilli) juvenile and adult mice by assessing the colony forming units (CFU) in lung homogenates at 4wpi. In juvenile mice, the average bacterial load increased (though not statistically significant) in the lungs of HFD fed infected mice compared to RD fed infected mice (Appendix A). However, in adult mice, the bacterial load in the lungs of HFD mice significantly increased (*p* ≤ 0.05) compared to RD fed mice (Appendix A). Auramine Rhodamine (A-R) staining of the lung sections showed the presence of *Mtb* in the lungs of both RD and HFD fed mice. In the infected juvenile mice, A-R staining demonstrated higher levels of *Mtb* in the lungs of HFD fed mice compared to RD mice. In infected adult mice, A-R staining showed higher *Mtb* load in the lungs of RD fed mice compared to juvenile mice, which further increased in HFD mice (Appendix A). Histological analysis of H&E sections of the lungs showed decreased alveolar space and increased septal thickening due to fibrosis, increased accumulation of lipid droplets and infiltrated immune cells such as macrophages, neutrophils, T cells and mast cells in the lungs during infection in both juvenile and adult mice irrespective of diet (Figure 2b). H&E-stained sections of *Mtb*-infected lungs also showed the presence of foamy macrophages. Compared to infected juvenile mice, infected adult mice showed increased pulmonary pathology (Figure 2b,c and Appendix A). In both infected juvenile and adult mice, increased lipid accumulation was observed in the lungs of HFD mice compared to RD mice (Appendix A). On the other hand, in infected adult mice, increased infiltrated immune cells were observed in RD fed mice compared to HFD mice (Figure 2b,c). The lungs of adult mice displayed reduced cytoplasmic protein levels compared to juvenile mice as indicated by eosin staining (Appendix A).

We also examined white adipose tissue (WAT) sections in both RD and HFD fed mice (juvenile and adult) by A-R staining and detected *Mtb* in the vicinity of lipid droplets and around the dead/dying adipocytes (Figure 3 and Appendix A). Despite this evidence of *Mtb* presence in WAT, we could not measure the bacterial loads in adipose tissue by CFU assessment because bacterial growth was very low in most of the samples 4 weeks after plating. This may be due to the presence of high amounts of lipids in WAT homogenates, which could prevent the active replication of *Mtb* on agar plates [27,28].

Histological sections of H&E-stained WAT showed the presence of infiltrated immune cells, loss of lipid droplets and fibrosis in infected mice. These pathological changes were more pronounced in adult mice compared with juvenile mice (Figure 4). As expected, whereas HFD increased the size of adipocytes/lipid droplets, *Mtb* infection resulted in a significant loss of fat cells in WAT in HFD mice compared to RD mice in both juvenile and adult groups (Figure 4).

Diet regulates adipose tissue physiology and immune signaling differently in juvenile and adult infected mice: Since we observed the presence of *Mtb* in the vicinity of lipid droplets associated with dying/dead adipocytes in adipose tissue in *Mtb* infected mice, we examined the effect of a HFD on adipose tissue pathophysiology during infection. The adipocyte functions are mainly regulated by its adipogenic signaling. We measured the levels of adipogenic markers in WAT of juvenile and adult mice infected with *Mtb* by immunoblot analysis (Appendix A). The levels of cleaved adiponectin (globular adiponectin fragment, 18–25 kd protein) increased in WAT in two of the five infected HFD-fed juvenile mice as compared to juvenile controls (not statistically significant). However, the levels of globular adiponectin significantly increased (*p* < 0.01) in WAT of infected RD-fed adult mice (Appendix A). We observed no significant difference in adiponectin levels (monomer or globular) between HFD-fed infected and control adult mice. In adult infected mice, the levels of globular domains of adiponectin in WAT of HFD-fed mice were significantly reduced (*p* ≤ 0.05) compared to RD-fed mice (Appendix A). Since the full-length adiponectin is known to regulate both adipogenesis and lipid oxidation [29,30], we analyzed the levels of PPAR-γ (peroxisome proliferator-activated receptor gamma) and PPAR-α, key regulators of adipogenesis and lipid oxidation, respectively (Appendix A). PPAR-γ levels were significantly decreased in both RD and HFD-fed infected juvenile mice compared to juvenile control mice (Appendix A). In adult mice, PPAR-γ levels decreased in both RD and HFD fed infected mice compared to control mice, but these changes did not reach significance (Appendix A). In adult mice, the levels of PPAR-α increased significantly in infected RD and HFD fed mice compared to adult control mice (Appendix A).

Histological analysis demonstrated increased infiltration of macrophages in adipose tissue in infected mice (Figure 4). Because activated macrophages secrete elastase, which cleaves adiponectin to form pro-inflammatory globular adiponectin fragments [31,32], we analyzed the effect of infection and diet on inflammatory signaling by measuring the levels of F4/80 (macrophage marker) and TNFα and IFNγ in the WAT of juvenile and adult mice fed different diets (Appendix A). We observed no significant change in F4/80 in WAT in juvenile mice. In the WAT of adult mice, F4/80 expression showed a significant increase in HFD-fed uninfected mice, which further increased in HFD-fed infected mice compared to adult control mice. In juvenile mice, there was a significant increase in inflammatory cytokine IFNγ protein levels in HFD-infected mice compared to RD-fed control mice, whereas there was no change in protein levels of TNFα in the WAT between uninfected and infected groups. Finally, adult mice showed a significant increase in TNFα levels in RD-fed infected mice but not in HFD-fed infected mice compared to adult control mice (Appendix A). This could be due to the increased levels of globular adiponectin fragment in infected RD fed adult mice compared to infected HFD fed mice (Appendix A). Our results indicate that in adult mice, a fat rich diet increases the levels of macrophage infiltration during infection but does not induce pro-inflammatory cytokine TNFα, whereas a carbohydrate rich RD induces proinflammatory status in the WAT during *Mtb* infection, which may be due to the increased levels of globular adiponectin in infected RD mice.

*Mtb* infection induces adipocyte lipolysis and apoptotic cell death in adult infected mice: It has been demonstrated that irregular ATGL induced lipolysis in adipocytes contributes to the development of insulin resistance [33]. We measured the levels of ATGL in infected juvenile and adult mice. The levels of P-ATGL in WAT significantly decreased in infected RD-fed mice and remained unchanged in infected HFD-fed juvenile mice compared to control mice (Appendix A). We also measured the levels of P-HSL and P-Perilipin, which are involved in the hydrolysis of lipids and degradation of lipid droplets in WAT. The levels of P-HSL significantly decreased (*p* < 0.05), however, the levels of P-Perilipin significantly increased (*p* < 0.05) in WAT of both infected RD-fed and HFD-fed juvenile mice compared to control mice (Appendix A). These data suggest that increased P-Perilipin may initiate basal lipolysis in adipocytes and that reduced levels of P-HSL inhibit the amplification of lipolysis in WAT during infection in juvenile mice [34].

In adult mice, immunoblot analysis showed that the P-ATGL levels significantly increased in WAT of infected HFD-fed mice (*p* < 0.01, respectively) compared to control mice (Figure 5a). The levels of P-ATGL in WAT of infected HFD-fed mice significantly increased (*p* < 0.01) compared to infected RD-fed mice (Figure 5a). The levels of P-HSL in WAT significantly increased in infected RD-fed (*p* < 0.01) and decreased in HFD-fed mice (*p* < 0.01) compared to control mice. Between the infected groups p-HSL significantly increased (*p* < 0.001) in infected RD-fed mice compared infected HFD-mice. The levels of P-Perilipin 1 significantly increased in infected RD-fed mice compared to control mice and infected HFD-fed mice (*p* < 0.001 and *p* < 0.05, respectively). Perilipin 1 levels did not change between the groups in adult mice (Figure 5a). These data suggest that p-HSL/p-Perilipin 1 associated lipolysis dominates in RD-fed mice, whereas ATGL induced lipolysis in HFD-fed mice in the loss of lipid droplets in adipose tissue in adult mice during *Mtb* infection.

Thus, *Mtb* infection can regulate lipolysis and degradation of lipid droplets via activation of different lipases in WAT irrespective of diet in adult mice. It has been demonstrated that ATGL-deficient mouse embryonic fibroblasts (MEFs) exhibited reduced p-AMPK and increased p-mTOR, which induces anti-apoptotic signaling pathway, whereas ATGL-overexpressing MEFs showed decreased apoptotic signaling [35]. On the other hand, activation of p-HSL/p-Perilipin 1 via TNFα demonstrated increased apoptosis. We analyzed the levels of P-AMPK, P-Akt and P-mTOR, which are major regulators of cell metabolic pathways, in the WAT of infected and uninfected juvenile and adult mice fed on different diets (Figure 5b and Appendix A). The data showed no significant difference in the levels of p-AMPK, p-Akt and p-mTOR between uninfected and *Mtb* infected juvenile mice fed either a HFD or RD (Appendix A). However, in adult mice, p-AMPKα levels significantly increased in WAT of HFD-fed infected mice as compared to adult controls (Figure 5b). The levels of P-Akt significantly decreased (*p* < 0.05) in RD-infected mice and further significantly decreased (*p* < 0.05) in HFD-infected mice compared to control mice (Figure 5b). Finally, the levels of P-mTOR were significantly reduced (*p* < 0.05) in HFD-infected mice compared to adult control mice (but were unchanged in RD infected mice). These data indicate that in adult mice the major energy signaling pathways altered during *Mtb* infection due to increased lipolysis in adipose tissue [35].

Histological sections of WAT showed significantly increased loss of adipocytes/cell death in *Mtb* infected mice compared to control mice (Figure 4). To examine whether the loss of adipocytes is due to necrosis or lipolysis induced apoptosis, we analyzed the levels of markers of necrosis (BNIP3) and apoptosis (cleaved caspase-3) in WAT (Figure 5c). Although the levels of BNIP3 did not change in infected mice compared to control mice, the levels of cleaved caspase 3 significantly increased in WAT in infected mice compared to control mice irrespective of the diet fed (Figure 5c). These data suggest that *Mtb* infection induces lipolysis and apoptosis in WAT in HFD-fed mice via the ATGL-mTOR pathway [35], whereas in RD-fed mice via TNFα-HSL-Perilipin 1 pathway [36,37].

Adipocyte dysfunction alters circulating lipid profiles in different age groups of infected mice fed on different diets: The altered metabolic and immune signaling and increased loss of lipid droplets in WAT during infection may regulate the serum lipid profile. Therefore, we measured the level of HDL, LDL/VLDL, total cholesterol, triglycerides, and fatty acids in the serum of juvenile and adult *Mtb* infected and control mice fed on RD and HFD at 30 DPI. The levels of HDL and total cholesterol significantly increased in the infected mice in both RD and HFD fed juvenile and adult mice. The levels of LDL/VLDL increased significantly (*p* < 0.05) in HFD-fed infected mice. The serum level of fatty acids did not differ between infected and uninfected groups in juvenile mice. In adult mice, the serum level of fatty acids increased significantly (*p* < 0.05) in HFD-fed infected mice as compared to adult control mice. Finally, juvenile mice fed on either diet showed a significant decrease (*p* < 0.01) in triglyceride levels upon *Mtb* infection, and adult mice showed a significant increase (*p* < 0.05) in triglyceride level in HFD-infected mice as compared to RD-fed control mice (Figure 6).

Impaired insulin signaling in the lungs of *Mtb* infected adult mice: Our data demonstrated that *Mtb* infected adult mice developed hyperglycemia and insulin resistance (based on the HOMA IR levels, Figure 1) irrespective of diet. To investigate whether systemic insulin resistance affected insulin signaling in the lungs, we analyzed expression levels of insulin receptors in the lungs by Western blotting (Figure 7). We found that the levels of insulin receptors were significantly reduced (*p* < 0.05) in the lungs of RD and HFD fed infected adult mice compared to control mice (RD fed uninfected mice). We next analyzed markers of insulin signaling pathway by assessing the levels of p-IRS1, p-Akt, p-GSK3b, and p-mTOR (Figure 7). Western blotting analysis demonstrated significantly reduced (*p* < 0.05) levels of p-IRS1, p-Akt and p-GSK3b in the lungs of infected (RD and HFD) mice compared to control mice. In addition, the levels of both mTOR and p-mTOR significantly decreased in the lungs of both RD (*p* < 0.05 and *p* < 0.01, respectively) and HFD (*p* < 0.01 and *p* < 0.005, respectively) fed infected adult mice compared to control mice. These data demonstrated impaired insulin signaling in the lungs of *Mtb* infected adult mice.

The data shown above (AKT/mTOR/GSK3b) also demonstrated that the biosynthetic pathways (e.g., producing proteins and glycogen) were inhibited, suggesting that there could be a depletion of energy resources in infected animals. Hence, we analyzed the levels of AMPK activation in the lungs by evaluating the levels of phosphorylated AMPK (Thr172) by Western blotting (Figure 8). Although the levels of AMPKα significantly increased (*p* < 0.01) in the lungs of both RD and HFD fed infected mice, the levels of p-AMPK significantly decreased (*p* < 0.01) in the lungs of infected mice compared to control mice. The levels of glycolytic enzyme hexokinase II (HK2) also significantly decreased (*p* ≤ 0.05) in the infected (RD and HFD) mice compared to control mice. These data suggest that AMPK activation and glycolysis are limited in the lungs during infection. Furthermore, because HK2 is the predominant isoform in insulin-sensitive tissues [38,39], its decreased level in the lungs further confirms the reduced insulin sensitivity in the lungs during infection in the adult mice.

*Mtb* infection affects lipid metabolism in the lungs of adult mice: In the absence of AMPK activation and glycolysis, the lungs may depend on lipids as their source of energy. Therefore, we analyzed lipid metabolic pathways by measuring the levels of some of the major enzymes (pyruvate dehydrogenase (PDH), ATP-citrate lyase (ACLY) and cytoplasmic acetyl-CoA synthetase (AceCS1)) involved in acetyl CoA biosynthesis, a main precursor of both lipid anabolism and catabolism (Figure 9). The levels of PDH and AceCS1 were significantly increased in RD (*p* < 0.05 and *p* < 0.01, respectively) and HFD (*p* < 0.05 and *p* < 0.005, respectively) fed infected mice compared to control mice. PDH catalyzes the conversion of pyruvate to acetyl Co-A, whereas AceCS1 catalyzes the conversion of acetate and CoA to acetyl-CoA, and thus increases the levels of acetyl-CoA. The levels of p-ACLY (catalytically active form of ACLY) were significantly decreased (*p* < 0.001) in the lungs of infected RD and HFD fed adult mice compared to control mice (Figure 9).

Next, we examined the role of acetyl Co-A in lipid metabolism (lipid biosynthesis vs. lipid oxidation). We analyzed the levels of enzymes involved in lipid biosynthesis (Acetyl-CoA carboxylase (ACC), fatty acid synthase (FASN) and sterol regulatory element binding protein-1 (SREBP1) in the lungs by Western blotting (Figure 9). ACC catalyzes the carboxylation of acetyl-CoA to malonyl-CoA [40,41]. This irreversible reaction is the committed step in fatty acid synthesis (FAS). FAS catalyzes the synthesis of saturated long chain fatty acids using acetyl CoA, malonyl CoA and NADPH [42]. SREBP1 is a transcription factor that regulates the genes involved in lipid biosynthesis [43]. The levels of ACC increased (though not significantly) and the levels of p-ACC (inhibitory form) significantly decreased (*p* < 0.001) in the infected (RD and HFD) mice compared to control mice (Figure 9), suggesting that the levels of malonyl-CoA increased in the lungs of infected mice. However, the levels of FASN did not increase and SREBP1 significantly decreased in RD and HFD (*p* ≤ 0.01 and *p*≤ 0.005, respectively) fed infected mice compared to control mice. These data suggest that although the levels of malonyl CoA and acetyl CoA increase during infection, lipid biosynthesis is inhibited in the lungs of adult infected mice, even those fed a HFD.

Next, we analyzed the levels of markers of lipid degradation and oxidation (p-perilipin and PPARα), lipases (phospho-HSL and ATGL), long-chain acyl-CoA synthetase (ACSL1) and CPT1 in the lungs by Western blotting (Figure 10). The levels of PPARα, a nuclear transcription factor involved in fatty acid oxidation [44,45] significantly increased (*p* < 0.05) in the lungs of infected mice compared to control mice. We also showed increased accumulation of lipid droplets in the lungs of infected mice. The degradation of lipid droplets is regulated by p-perilipin and activation of lipases [46]. Western blotting analysis demonstrated increased levels of p-perilipin and ATGL and significantly reduced P-HSL (*p*< 0.05) in the lungs of infected mice compared to control mice irrespective of the diet fed. ACSL1 catalyzes the ligation of the fatty acid to CoA to form fatty acyl-CoA, which is transported to mitochondria via CPT1 for β-oxidation to release energy in the form of ATP [47]. Both ACSL1 and CPT1A levels significantly increased (*p* < 0.05) in the lungs of infected mice compared to control mice irrespective of the diet fed. These data suggest that lipid β-oxidation is prominent in the lungs of infected mice and may provide the essential ATP for the cellular activity during infection.

## 4. Discussion

The epidemiological link between diabetes and increased risk of active TB disease is well documented. This study examined the corollary of the link between diabetes and TB disease and whether: (i) a low grade *Mtb* infection induces hyperglycemia and insulin resistance; (ii) a high-fat diet during *Mtb* infection increases the risk of insulin resistance; (iii) the development of metabolic syndrome in an *Mtb* infected host is age dependent; and (iv) adipocytes/adipose tissue contribute to the development of insulin resistance during *Mtb* infection. Specifically, to examine the effect of diet, age, and *Mtb* infection in the pathogenesis of metabolic syndrome, we used C57BL/6 murine models fed either a HFD or RD and infected with low levels of *Mtb* (10^6^) during the juvenile (6 weeks old) and young adult (10 weeks old) age (Appendix A). Our study revealed that adult mice are more susceptible to developing hyperglycemia and insulin resistance and that a HFD increases the severity of insulin resistance in *Mtb* infected mice (Figure 1). These results provide evidence that a low level *Mtb* infection and persistence, such as may occur with latent TB infection which affects a quarter of the world’s population, may contribute to the pathogenesis of diabetes.

Although the serum levels of triglycerides were higher in RD and HFD fed control juvenile mice compared to their respective diet fed control adult mice, the levels of other lipids (such as LDL, total cholesterol, and fatty acids) were not significantly changed. However, the levels of HDL significantly increased in the *Mtb*-infected RD and HFD fed juvenile and adult mice (Figure 6). Increased levels of circulatory HDL have been shown to increase insulin secretion and reduce blood glucose levels [48]. In addition to HDL, circulatory fatty acids also induce insulin secretion [49,50]. Our data showed that insulin secretion induced by circulatory lipids (HDL and fatty acids) did not reduce the levels of blood glucose in infected adult mice, suggesting that lipid-storing adipocytes may be compromised and dysfunctional in regulating blood glucose levels during infection. Furthermore, RD and HFD fed adult mice showed increased body weight and expanded visceral adipose tissue compared to their respective diet fed juvenile mice, suggesting that an age dependent increase in body fat may play a role in pulmonary pathology and development of insulin resistance.

Earlier, we showed that even though the lungs are the site of *Mtb* infection and pathogenesis in aerosol-infected mice, *Mtb* infection directly and indirectly affects adipocyte/fat cell physiology [23,51]. In particular, using a transgenic fat-modulatable murine TB model we demonstrated that adipocytes/adipose tissue serve as a reservoir for *Mtb* and that an acute loss of fat cells increases pulmonary pathology and bacterial load and decreases immune cell activation via elevated accumulation of lipid droplets in the lungs of *Mtb* infected mice [23]. Since we had previously found that an acute loss of fat cells increases pulmonary pathology, we examined the effect of diet induced adiposity on pulmonary pathology in *Mtb* infected mice in this current study. Diets rich in fat and carbohydrate differently regulate adipocyte physiology [52]. Both pathogenic adipocytes and diets (high-fat and high-carbohydrate diets) regulate whole body energy metabolism and the pathogenesis of insulin resistance and diabetes [53,54]. In case of juvenile mice, a HFD was fed only for a short time (during early age), which slightly increased adiposity (based on the size of the adipocytes and weight gain) compared to RD fed mice. This slight increase in adiposity in HFD-fed juvenile mice had no significant effect on the basal glucose and insulin levels (or HOMA IR) in both infected and uninfected mice. Interestingly, however, in adult mice a HFD significantly increased adiposity and *Mtb* infection caused even higher levels of insulin resistance, as evident with the HOMA-IR data (both RD and HFD fed infected mice compared to controls) (Figure 1). The observed significant increase in the levels of TNFα in adipose tissue of infected RD adult mice (Appendix A) may induce a proinflammatory status that could contribute to the development of systemic insulin resistance [55]. The infected RD-fed and HFD-fed mice demonstrated activation of different lipase pathways (TNF/pHSL vs. ATGL/mToR), which ultimately caused extensive lipolysis and apoptosis in WAT and increased lipid accumulation in other organs. Importantly, *Mtb* infected adult mice developed hyperglycemia and insulin resistance irrespective of diet. In addition, *Mtb* infected mice displayed lower body weight compared to their respective diet fed uninfected mice. It is interesting to note that the average HOMA-IR levels did not significantly differ between the RD fed infected adult mice and the HFD fed uninfected adult mice, although the body weights of RD fed infected adult mice were significantly lower (*p* ≤ 0.01) compared to HFD fed uninfected adult mice. Because these data suggest that the development of insulin resistance during *Mtb* infection may not be entirely due to HFD or adiposity, this may explain why latent TB infected individuals in TB endemic regions, who are mostly not obese and do not have higher BMI, are more prone to developing insulin resistance and diabetes [56,57].

We demonstrated an increased loss of lipid droplets and cell death in WAT of infected adult mice compared to uninfected mice irrespective of the diet fed. Lipolysis of adipocytes is mainly triggered by the enzymes ATGL and p-HSL [58,59]. Lipolysis and adipogenesis are tightly regulated in adipose tissue and directly correlated to insulin levels. In this study we demonstrated that the loss of lipid droplets is mainly due to the overexpression and activation of ATGL and p-HSL in WAT in HFD-fed and RD-fed mice, respectively, during infection (Figure 5). TNFα, which is induced by *Mtb* infection in WAT in RD-fed mice might have caused p-HSL regulated loss of lipid droplets [36], whereas HFD likely contributed to the increased loss of lipid droplets via ATGL [60]. It has been shown that impaired regulation of adipocyte lipolysis by ATGL contributes to the development and exacerbation of insulin resistance by increasing circulatory lipid levels and increasing lipid and proinflammatory immune cell infiltration into other organs and metabolic tissues [33]. Thus, the development of insulin resistance in *Mtb* infected adult mice may be due to the irregular lipolysis and apoptosis in adipocytes and altered metabolic signaling in other organs, including the lungs, during infection.

In prior studies, we have shown that increased loss of fat cells correlated with elevated levels of lipid droplets in other organs, including the heart and lungs [23,61]. Indeed, we detected increased lipid droplets in the lungs of infected adult mice (Figure 2 and Appendix A), leading us to hypothesize that the lungs are actively breaking down intracellular lipids to reduce their lipotoxic effects and obtain energy. In support of this, we found that the lung cells activated a lipid β-oxidation process during infection as demonstrated by increased levels of PPARα regulated proteins involved in lipid β-oxidation (Figure 10) and reduced levels of SREBP regulated proteins involved in lipid biosynthesis (Figure 9). We also showed a significantly reduced level of protein biosynthesis (mTOR signaling) (Figure 7), lipid biosynthesis (SREBP signaling) (Figure 9) and glucose uptake (AMPK activation) and glycolysis (HK2) (Figure 8) in the lungs during infection in adult mice irrespective of diet. Together, these data indicate that the lungs shut down the major energy signaling pathways, including insulin signaling, and become non-responsive to circulating glucose and insulin levels during persistent *Mtb* infection.

Our study was focused on analyzing the changes in the circulatory lipid, glucose, and insulin levels and energy signaling pathways and correlating these data to adipose tissue pathophysiology, pulmonary *Mtb* burden and pathology, and insulin resistance in acute *Mtb* infected mice. The major limitation of our study is that after the acute *Mtb* infection (above 30 dpi), the physiology of adipose tissue may improve due to the resolution of inflammation, which may increase insulin sensitivity at least in RD fed infected mice; thus, our study does not reveal whether *Mtb* infection induced insulin resistance in adult mice is transient or pathogenic to T2DM progression.

Our results show that *Mtb* infection in adult mice induces hyperglycemia, hyperinsulinemia, and insulin resistance even in mice fed an RD. Furthermore, although HFD fed *Mtb* infected adult mice displayed lower body weight compared to HFD fed adult uninfected mice, they showed significantly greater levels of insulin resistance. These findings may underpin the clinical observation that non-obese/non-high-BMI individuals with LTBI are more likely to develop hyperglycemia and diabetes compared to those without *Mtb* infection [62]. Our data also suggest that an increased lipolysis/acute loss of adipocytes in adipose tissue and altered lipid metabolism in other organs including the lungs due to an increased accumulation of intracellular lipids may drive the pathogenesis of insulin resistance during *Mtb* infection [63,64]. In addition, the proinflammatory adipose tissue may contribute to systemic insulin resistance in RD-fed adult *Mtb* infected mice. Therefore, we conclude that the physiology of deregulated adipocytes is associated with development of insulin resistance during *Mtb* infection. Our study warrants further investigation of the role of adipose tissue and loss/gain of fat mass in clinical subjects to understand the link between *Mtb* infection and metabolic syndrome. 

## Figures and Tables

**Figure 1 jcm-11-01646-f001:**
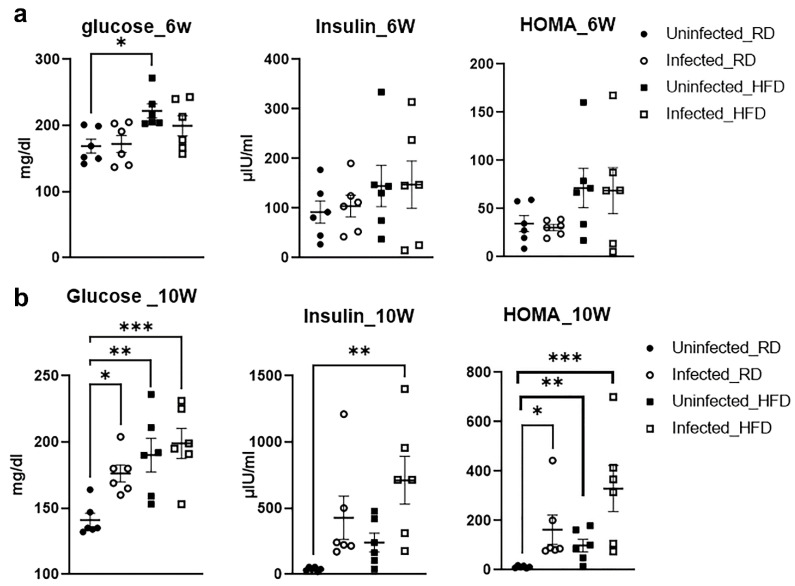
Development of insulin resistance in H37Rv *Mtb* infected mice (30 DPI) is age dependent (*n* = 5/group). (**a**) Serum levels of glucose, insulin and HOMA-IR in juvenile mice measured by ELISA. (**b**) Serum levels of glucose, insulin and HOMA-IR in adult mice measured by ELISA. The error bars represent standard error of the mean. * *p* < 0.05, ** *p* < 0.01, and *** *p* < 0.001, comparison between the indicated groups.

**Figure 2 jcm-11-01646-f002:**
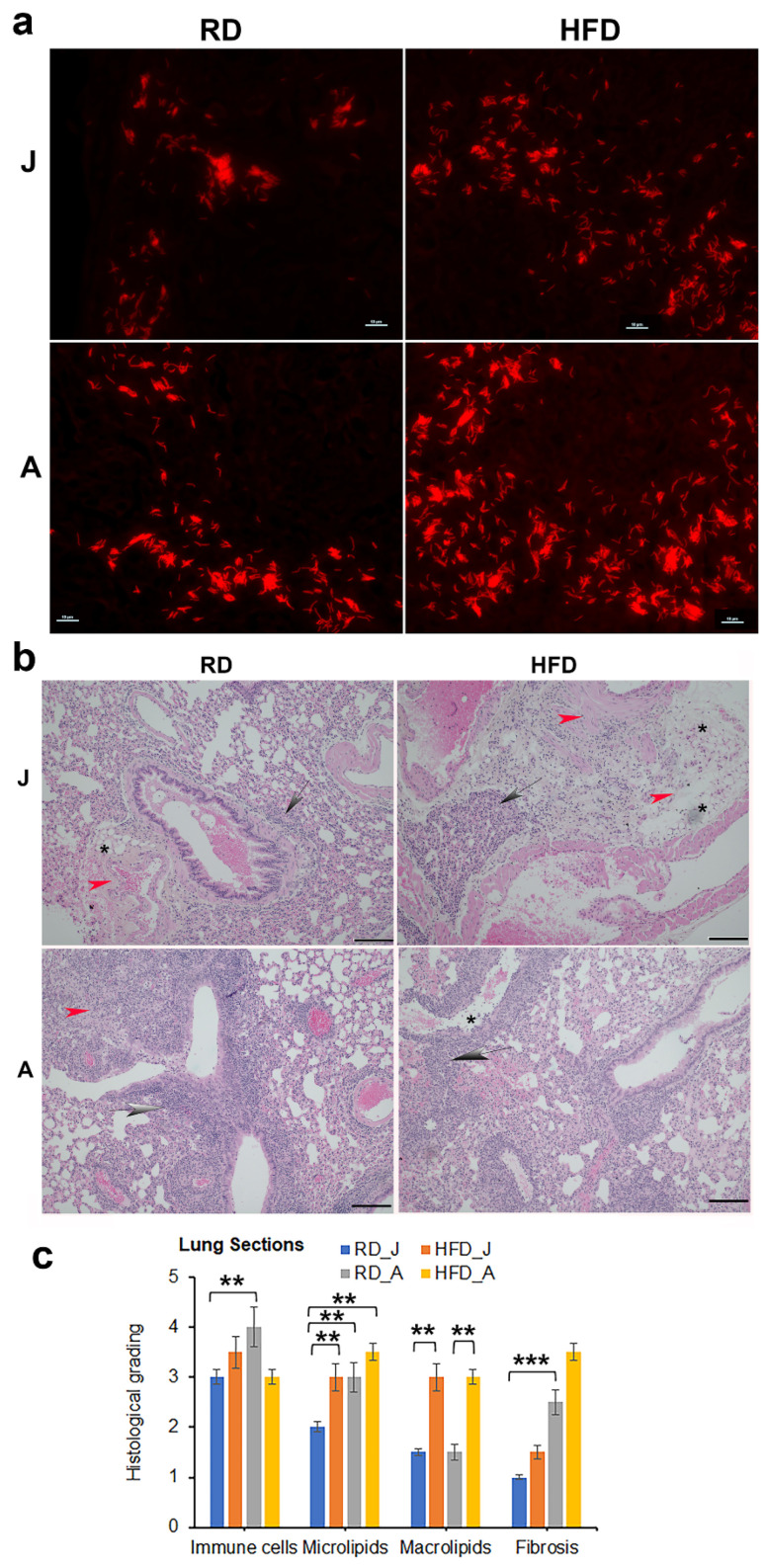
Diets differently regulate *Mtb* load in the lungs in juvenile and adult mice (30 DPI, *n* = 5/group). (**a**) Auramine-rhodamine (AR) staining (fluorescent red rods) demonstrated the presence of *Mtb* in the lung sections of H37Rv infected juvenile and adult mice (×40 magnification). (**b**) Histological analysis of the lungs demonstrated increased lung pathology (infiltrated immune cells [grey arrows]) and decreased alveolar space in *Mtb*-infected adult mice compared to juvenile mice fed a RD or HFD. The presence of lipid droplets [black stars] and fibrosis [red arrowhead] are shown in the images (×20 magnification, scale bar = 100 μm. (**c**) Histological grading of lung pathology was carried out according to experimental groups and classified in terms of infiltrated immune cells, microlipids, macrolipids and fibrosis. Each class was graded on a 6-point scale ranging from 0 to 5 as discussed in Materials and Methods. Values plotted are mean ± standard error (SE) from *n* = 5. The error bars represent the standard error of the mean. * *p <* 0.05, ** *p* < 0.01, and *** *p* < 0.001, between the indicated groups.

**Figure 3 jcm-11-01646-f003:**
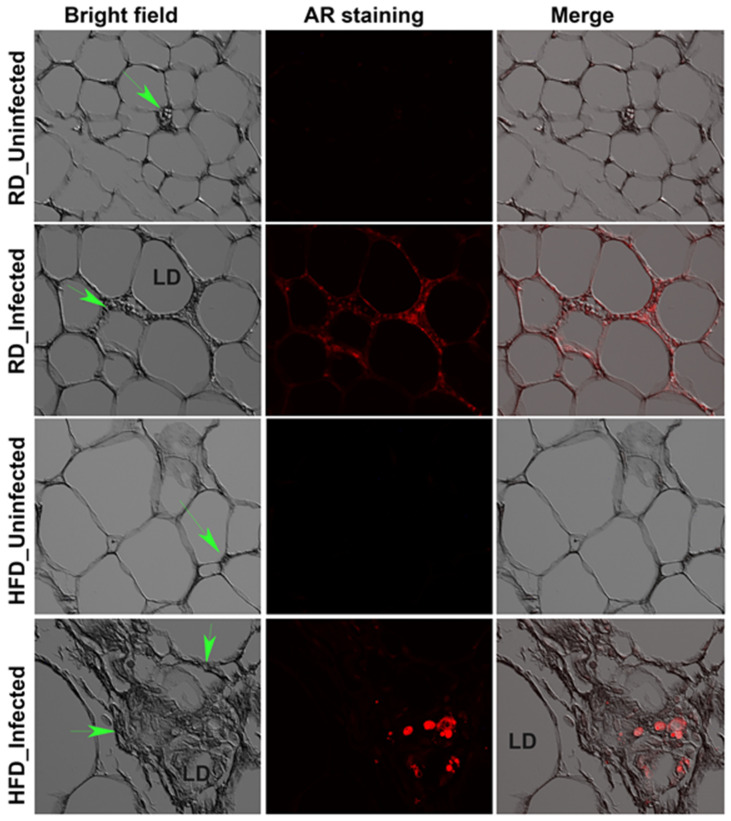
*Mtb* is present in the visceral fat sections of RD and HFD fed adult mice aerosol infected with H37Rv as demonstrated by AR staining (fluorescent red dots) at 30 dpi (×40 magnification). Bright field images of adipose tissue demonstrate the change in adipocyte size between RD and HFD fed mice and the merged images show the presence of *Mtb* (red) around dying adipocytes. Immune cells (green arrow).

**Figure 4 jcm-11-01646-f004:**
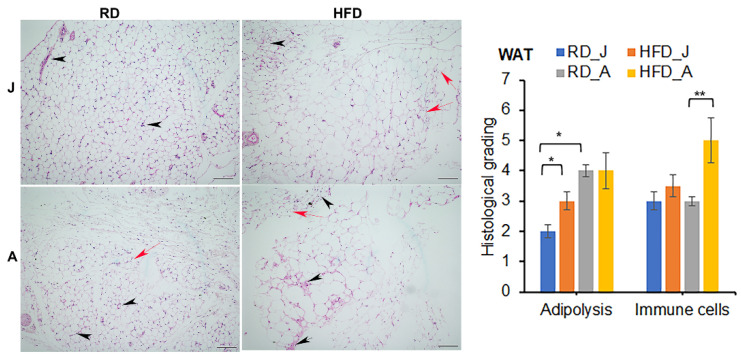
*Mtb* infection alters adipose tissue pathology by causing infiltration of immune cells into adipose tissue (black arrowheads), and loss of adipocytes (red arrows, smaller adipocytes) in juvenile and adult aerosol *Mtb* infected mice fed RD and HFD, as shown by H&E staining. Adult infected mice showed increased adipose tissue pathology compared to juvenile infected mice as demonstrated by H&E staining at 30 dpi. (×10 magnification, scale bar = 100 μm). Each class was graded on a 5-point scale ranging from 0 to 5 as discussed in Materials and Methods. Values plotted are mean ± standard error (SE) from *n* = 5. The error bars represent the standard error of the mean. * *p* < 0.05, and ** *p* < 0.01, between the indicated groups.

**Figure 5 jcm-11-01646-f005:**
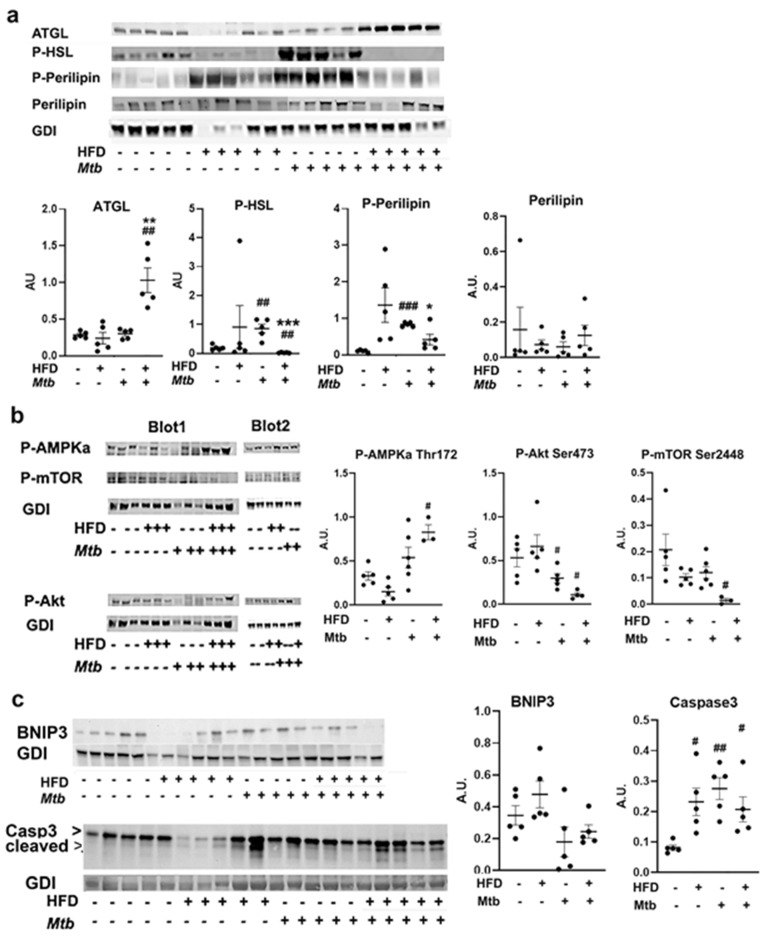
*Mtb* infection induces adipocyte lipolysis and apoptotic cell death in adult infected mice (**a**) Immunoblot analysis of lipid metabolism proteins (ATGL, P-HSL, P-Perilipin Ser 522, Perilipin) in the WAT lysates of infected and uninfected adult mice. (**b**) Immunoblot analysis of cell metabolic pathway proteins (P-AMPKα Thr 172, P-Akt Ser473, P-mTOR Ser 2448) in the WAT lysates of infected and uninfected adult mice. (**c**) Immunoblot analysis of cell death pathway proteins, necrosis (BNIP3) and apoptosis (Caspase3) in the WAT lysates of infected and uninfected adult mice. GDI was used as loading control. Fold changes in the protein levels were normalized to GDI expression and are represented as a dot plot. The error bars represent standard error of the mean. # *p* < 0.05, ## *p* < 0.01, and ### *p* < 0.001, compared to uninfected RD mice, * *p* < 0.05, ** *p* < 0.01 and *** *p* < 0.001 compared to infected RD mice from *n* = 5/group.

**Figure 6 jcm-11-01646-f006:**
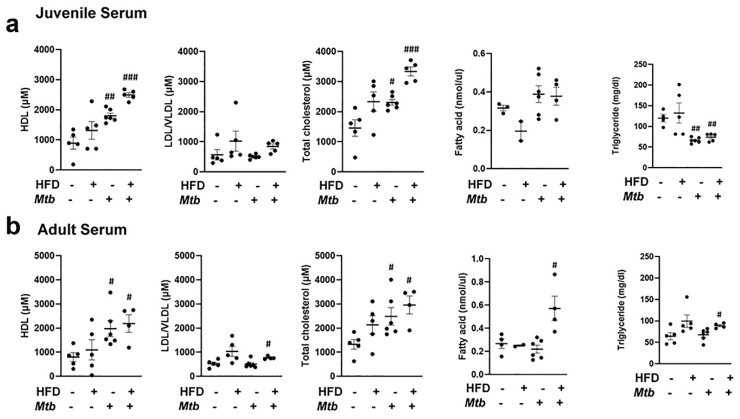
Changes in the circulatory levels of triglycerides, fatty acid, and cholesterol in different age group mice with infection fed on different diet at 30 DPI (*n* = 5/group). (**a**) Serum levels of HDL, LDL/VLDL, total cholesterol. fatty acid and triglyceride in juvenile mice measured by ELISA. (**b**) Serum levels of HDL, LDL/VLDL, total cholesterol. fatty acid and triglyceride in adult mice measured by ELISA. The error bars represent standard error of the mean. # *p* < 0.05, ## *p* < 0.01, and ### *p* < 0.001, compared to uninfected RD mice.

**Figure 7 jcm-11-01646-f007:**
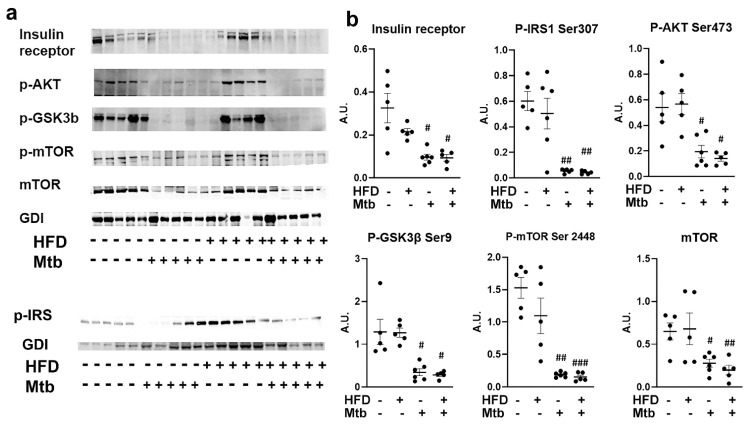
Impaired Insulin signaling in the lungs of *Mtb* infected adult mice (**a**) Immunoblot analysis of insulin receptor and insulin signaling proteins (p-IRS1 Ser307, p-AKT Ser473, p-GSK3β Ser9, p-mTOR Ser2448, mTOR) in the lung lysates of infected and uninfected adult mice. GDI was used as loading control. (**b**) Fold changes in the protein levels were normalized to GDI expression and are represented as a dot plot. The error bars represent standard error of the mean. # *p* < 0.05, ## *p* < 0.01, and ### *p* < 0.001, compared to uninfected RD mice (*n* = 5/group).

**Figure 8 jcm-11-01646-f008:**
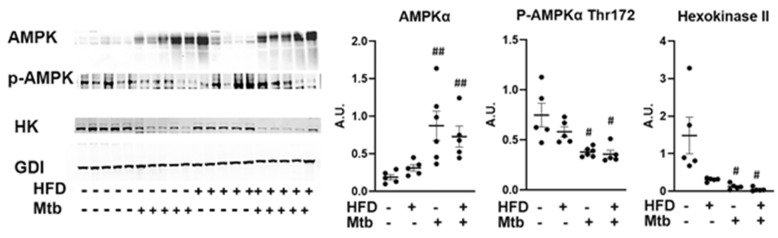
Changes in AMPK signaling in the lungs of *Mtb* infected adult mice. Immunoblot analysis of AMPK pathway proteins (AMPKα, p-AMPKα) and glycolytic enzyme (Hexokinase II) in the lung lysates of infected and uninfected adult mice. GDI was used as loading control. Fold changes in the protein levels were normalized to GDI expression and are represented as a dot plot. The error bars represent standard error of the mean. # *p* < 0.05, ## *p* < 0.01, compared to uninfected RD mice (*n* = 5/group).

**Figure 9 jcm-11-01646-f009:**
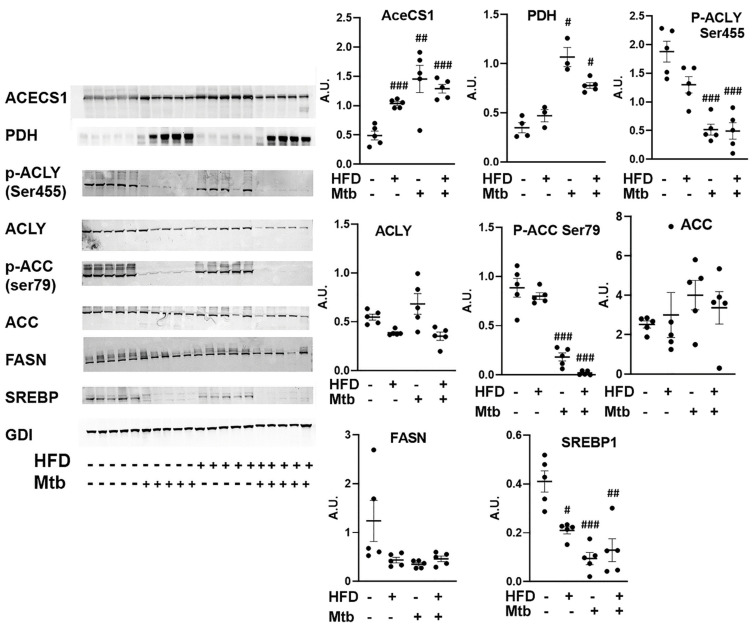
*Mtb* infection affects lipid metabolism in the lungs of adult mice. Immunoblot analysis of lipid metabolism pathway proteins (AceCS1, PDH, p-ACLY Ser455, ACLY, p-ACC Ser79, ACC, FASN, SREBP1) in the lung lysates of infected and uninfected adult mice. GDI was used as loading control. Fold changes in the protein levels were normalized to GDI expression and are represented as a dot plot. The error bars represent standard error of the mean. # *p* < 0.05, ## *p* < 0.01, and ### *p* < 0.001, compared to uninfected RD mice (*n* = 5/group).

**Figure 10 jcm-11-01646-f010:**
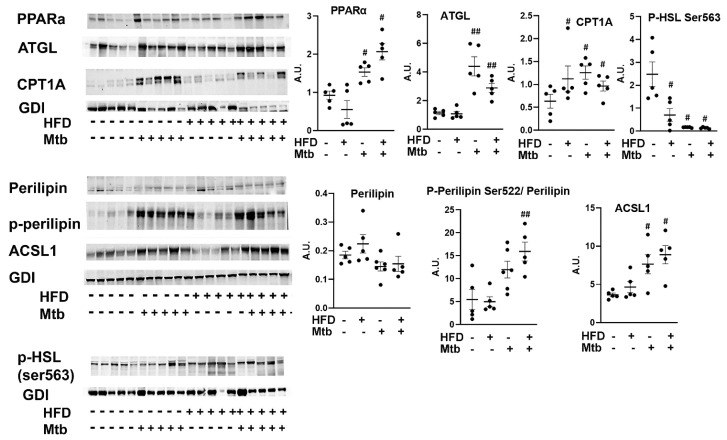
*Mtb* infection affects lipid metabolism in the lungs of adult mice. Immunoblot analysis of the levels of markers of lipid degradation and oxidation (PPARα, p-Perilipin Ser522, Perilipin), lipases (p-HSL and ATGL), long-chain acyl-CoA synthetase (ACSL1) and CPT1 in lung lysates of infected and uninfected adult mice. GDI was used as loading control. Fold changes in the protein levels were normalized to GDI expression and are represented as a dot plot. The error bars represent standard error of the mean. # *p* < 0.05, and ## *p* < 0.01 compared to uninfected RD mice (*n* = 5/group).

## Data Availability

All the data included in the manuscript.

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
