# Peer review of "Host Metabolic Changes during Mycobacterium Tuberculosis Infection Cause Insulin Resistance in Adult Mice"

_jcm, 2022, doi:10.3390/jcm11061646_

Round 1
Reviewer 1 Report
This manuscript evaluates the metabolic changes in lung and adipose tissue after Mtb infection in juvenile and adult mice, that were fed either regular diet (RD) or high fat diet (HFD). This work demonstrates that adult mice are more susceptible to metabolic syndrome after Mtb infection using RD or HFD. Unfortunately, CFU data form WAT was not possible, but Mtb presence in WAT was demonstrated by imaging; in addition, lesions produce by Mtb infection was reported in WAT. Therefore, lipid profiles measured in serum show higher levels of total cholesterol in HFD than RD as was expected. Interestingly, only Mtb infection increase similar levels of HDL and total cholesterol in RD than levels in HFD. Furthermore, in this works several keys molecules in metabolism were evaluated in WAT and lung, demonstrating that Mtb infection decrease glucose metabolism through insulin receptor in the lungs. Finally, lipogenesis molecules were decreased, and an increment of lipid oxidation markers were observed in lungs and WAT, suggesting that lipids could be the main energetic source during Mtb infection.
Major Comments
There were not any major comments
Minor Comments
- In page 4 the text indicates that “basal insulin levels were higher in both Mtb infected RD and HFD fed mice, but the graph only show significance in HFD.
- In the text Figure 1 has been referenced as 1a or 1b, but the figure doesn’t indicate the letters.
- In Figure 2, Shouldn’t be better a ZN to show Mtb in the lung? because of the tome post infection, a lot more bacteria are expected in the lung.
- In Figure 2 the resolution of all the images was very low and reviewer was not able appreciate their conclusions.
- In page 5 indicate that lipid droplets were increased in figure 2b, due to the image resolution and not markers were added there is vague to describe them as lipid droplets.
- In figure 3, green arrows are signaling macrophages; however, when an adipocyte die have been reported the recruitment of several immune cells. Was any marker used to define those cells as macrophages?
- In page 8 the text describes a significant increase of IFN γ in HFD, but the graph shows those levels decreased (Fig S4).
- The correct identification of perilipin as perilipin-1 is necessary due to the different molecules included in this family which have different function and regulations.
Author Response
We thank the reviewers for their evaluation and comments. As per the suggestions of the reviewers, we substantially revised the manuscript, including additional Figures and references. By doing so, we have addressed the reviewers’ comments and concerns, further strengthening our work and clarifying our results. We also identified a few mistakes during the revision, which we have addressed in the revised manuscript. We believe that by addressing the reviewers’ comments and including additional data we have significantly improved this manuscript and that it is now suitable for publication in the Journal of Clinical Medicine. The changes made to the manuscript are marked as suggested by the editorial office manager. In the event of publication, we assign copyright in accordance with the instructions to authors.
A detailed response to all the reviewers' suggested modifications and corrections follows. The reviewers’ comments are in italic font 10 text.
Reviewer #1 Comments and Author Responses
Comment: In page 4 the text indicates that “basal insulin levels were higher in both Mtb infected RD and HFD fed mice, but the graph only show significance in HFD.
Response: The basal insulin levels increased in both Mtb infected RD and HFD mice; however, a significant change was observed only in HFD-fed mice. We apologize for this confusing statement. In the revised manuscript (line 204), we revised the sentence as “basal insulin levels were significantly higher in Mtb infected HFD fed mice (p≤ 0.01) compared to control mice”.
Comment: 2. In the text Figure 1 has been referenced as 1a or 1b, but the figure doesn’t indicate the letters.
Response: We included the letters a and b in Figure.1.
Comment: In Figure 2, Shouldn’t be better a ZN to show Mtb in the lung? because of the tome post infection, a lot more bacteria are expected in the lung.
Response: We apologize for the low resolution of Figure 2. At 30 days post-infection we observed a moderate to a significant number of Mtb in the lungs depending on the groups as shown by the Auromine Rhodamine (AR) staining. Our previous studies did not show any difference between Mtb loads detected by ZN staining vs. AR staining. Moreover, it is difficult to identify Mtb in adipose tissue by ZN staining, since the blood cells also show pink coloration in adipose tissue. In the revised manuscript, we included better Figure 2 with higher resolution to show the presence of Mtb in the lungs.
Comment: In Figure 2 the resolution of all the images was very low and reviewer was not able appreciate their conclusions.
Response: This may be due to the change in resolution during the conversion between the Tif file and Word/PDF. In the revised manuscript, we included a better Figure 2 with a higher resolution. In addition, we will upload high-resolution figures separately.
Comment: In page 5 indicate that lipid droplets were increased in figure 2b, due to the image resolution and not markers were added there is vague to describe them as lipid droplets.
Response: We added a new supplemental figure (Fig. S2d) with higher resolution and magnification to show the presence of micro and macro lipid droplets and foamy macrophages in the lungs.
Comment: In figure 3, green arrows are signaling macrophages; however, when an adipocyte die have been reported the recruitment of several immune cells. Was any marker used to define those cells as macrophages?
Response: We changed the figure 3 legend in the revised manuscript. The green arrow indicates immune cells (not just macrophages).
Comment: In page 8 the text describes a significant increase of IFN γ in HFD, but the graph shows those levels decreased (Fig S4).
Response: We apologize for this typo. In the revised manuscript we corrected the text to say “decreased”.
Comment: The correct identification of perilipin as perilipin-1 is necessary due to the different molecules included in this family which have different functions and regulations.
Response: We agree with the reviewer. In the revised manuscript we refer to perilipin as perilipin-1 in the Materials and Methods section.
Reviewer 2 Report
Comments for authors
This study (Host Metabolic Changes During Mycobacterium tuberculosis Infection Cause Insulin Resistance in Adult Mice) used mice infection in investigating the characteristics of MTB infection in insulin resistance and the role of adipose tissue in pulmonary pathology upon MTB infection. The mechanisms related to lipid metabolism and insulin resistance is extensively evaluated in lung tissues and WAT. This is an interesting issue that deserves investigation. My comments are as below
- If my understanding is correct, the juvenile mice (6wk) received RD/HFD for 2 weeks but the adult mice (10wks) received RD/HFD received 6 weeks of RD/HFD. Why the juvenile mice and adult mice did not receive the same duration of HFD. The impact of aging in lipid metabolism and insulin resistance can be caused by the different HFD duration.
- Result, paragraph 1, the infected adult mice showed a significant decrease (2.2 ±5g) in their weight. What’s the weight change in RD and HFD fed adult mice respectively?
- It appears that RD actually has higher carbohydrate calories, please explain why HFD can induce higher glucose level than RD in uninfected mice.
- As insulin resistance is one of the key measurements in this study, please provide detail of HOMAIR measurement in the section of methods
- Please label (a) and (b) in figure 1.
- Figure 1b, the glucose level of infected_HFD should be compared with uninfected_HFD, not uninfected_RD. The rule also applies to insulin and HOMA.
- All the figures have low resolution. Please revise the figures. The resolution of Figure 2 is low, I can hardly see anything in Fig. 2a. The detailed histological feature of lung tissue cannot be identified in Fig 2b.
- As ATGL contributes to the development of insulin resistance, we assume that ATGL and HOMA should follow the same direction. In Fig 1, HOMA (insulin resistance) is not related to MTB infection but related to HFD in juvenile mice. Why can we see the difference in ATGL after MTB infection in Fig S3a?
- In Fig S4, the responses of IFN-g and TNF-a to diet and MTB appear different between juvenile and adult mice. What’s the possible mechanism to explain the differences between juvenile and adult mice.
- In Fig 1, in uninfected adult mice, HFD is associated with an increase in HOMA. However, in Fig 5, there is no difference in ATGL between RD and HFD mice uninfected to MTB. What is the explanation?
- In Figure 7~10, the authors investigated the expression of markers related to lipid metabolism in lung tissue from adult mice. I would recommend authors provide the results in juveniles to confirm the role of aging in MTB-related lipid metabolism in lung tissue.
- The expression of ATGL/P-HSL/P-perilipin in lung tissue and WAT in adult mice seems does not follow a similar pattern (Fig 5 and Fig 10). Can the authors explain the possible differences between lung tissue and WAT in lipid metabolism?
- To confirm the finding, I would recommend authors to confirm their findings in the levels of RNA expression in key markers related to lipid metabolisms, such as PPARa, ATGL, etc.
- In the final section, the authors speculated that weight loss in MTB individuals can be a risk factor for developing metabolic syndrome and diabetes. However, the correlation between weight loss and lipid metabolism and insulin resistance is not well investigated in mice and humans. I do not think this speculation is appropriate and should be removed.
Author Response
We thank the reviewers for their evaluation and comments. As per the suggestions of the reviewers, we substantially revised the manuscript, including additional Figures and references. By doing so, we have addressed the reviewers’ comments and concerns, further strengthening our work and clarifying our results. We also identified a few mistakes during the revision, which we have addressed in the revised manuscript. We believe that by addressing the reviewers’ comments and including additional data we have significantly improved this manuscript and that it is now suitable for publication in the Journal of Clinical Medicine. The changes made to the manuscript are marked as suggested by the editorial office manager. In the event of publication, we assign copyright in accordance with the instructions to authors.
A detailed response to all the reviewers' suggested modifications and corrections follows. The reviewers’ comments are in italic font 10 text.
Reviewer# 2’s Comments and Authors’ Response
Comment: If my understanding is correct, the juvenile mice (6wk) received RD/HFD for 2 weeks but the adult mice (10wks) received RD/HFD received 6 weeks of RD/HFD. Why the juvenile mice and adult mice did not receive the same duration of HFD. The impact of aging in lipid metabolism and insulin resistance can be caused by the different HFD duration.
Response: The weaning age of mice is approximately 25 days. We did not want to alter the diet during the weaning age (which represents babies). We were interested in studying the effect of diets in the infected juvenile and adult mice (that represent children and adults in patients and not babies/toddlers). Therefore, we started the diet treatment after 4 weeks of age. The juvenile group received only two weeks of different diets before infection. However, they were continued on their respective diets for 4 more weeks after infection. It is known that the duration of HFD treatment alters lipid metabolism and insulin resistance. However, our current study indicated that Mtb infection can also alter host lipid metabolism, which can be age-dependent.
Comment: Result, paragraph 1, the infected adult mice showed a significant decrease (2.2 ±5g) in their weight. What’s the weight change in RD and HFD fed adult mice respectively?
Response: In the revised manuscript, we provide the weights of mice with and without infection (both for RD- and HFD-fed mice). Mice fed on HFD showed increased (p≤0.01) body weight compared to RD fed mice at 6 weeks of age (19.8 ± 1.8g vs 17.5 ± 1.1g) and 10 weeks of age (28 ± 1.6g vs 23 ± 2g) (data not shown). RD and HFD fed uninfected adult mice showed significantly increased body weight (5.5 ± 0.9g and 8.2 ± 0.2 respectively; p≤0.01) and expanded visceral adipose tissue (1.5-fold and 2.5-fold, respectively; p≤0.01) compared to their respective diet-fed uninfected juvenile mice (data not shown). At 4 weeks post-infection (wpi), Mtb infected juvenile mice showed no significant differences in their body weights compared to their respective diet-fed uninfected mice (RD 23.2 ± 2.0g and HFD 27.6 ± 1.7g). However, the infected adult mice showed a significant decrease (p≤0.05) in their body weights compared to their respective diet-fed uninfected groups (uninfected RD vs infected RD (26.2 ± 1.7g vs 22.5 ± 1.8g) and uninfected HFD vs infected HFD (33.7 ± 2.2g vs 28.9 ± 1.9g) at 4 wpi.
Comment: It appears that RD actually has higher carbohydrate calories, please explain why HFD can induce higher glucose level than RD in uninfected mice.
Response: Yes, it is correct that RD has higher carbohydrate calories and HFD has higher fat calories. The increased basal glucose in the blood depends on two mechanisms: (i) impaired insulin levels and (ii) impaired insulin sensitivity. In uninfected RD-fed mice, insulin sensitivity and insulin levels are normal. However, in HFD-fed mice, increased circulating lipids can elevate abnormal insulin levels, which could cause insulin resistance and lead to increased glucose levels in the blood [PMCID: PMC5398492]. In addition, HFD increases lipid accumulation in the organs/cells, causing a shift in energy metabolism towards lipid catabolism. Furthermore, under this condition, the uptake of glucose is partially inhibited in the organs/cells to avoid lipid toxicity.
Comment: As insulin resistance is one of the key measurements in this study, please provide detail of HOMAIR measurement in the section of methods
Response: In the revised manuscript, we provided the details of HOMAIR measurement in the section of methods. We also rectified a small mistake in presenting the units (Y-axis) in Fig 1 (HOMA-IR) and hence uploaded revised Figure 1 in the revised manuscript. The new figure does not change the results that we previously presented.
Comment: Please label (a) and (b) in figure 1.
Response: In the revised manuscript, we label (a) and (b) in figure 1.
Comment: Figure 1b, the glucose level of infected_HFD should be compared with uninfected_HFD, not uninfected_RD.
Response: Our study investigated the effect of high-fat diet and Mtb infection on the levels of basal glucose and insulin in comparison to the control RD. Therefore, we compared the levels of glucose and insulin in all the groups to RD uninfected mice. As suggested by the reviewer, in the revised manuscript we also include information on any changes in the levels of glucose, insulin and HOMA as compared between uninfected and infected HFD-fed mice.
Comment: All the figures have low resolution. Please revise the figures. The resolution of Figure 2 is low, I can hardly see anything in Fig. 2a. The detailed histological feature of lung tissue cannot be identified in Fig 2b.
Response: We apologize for the low resolution of Figure 2. This may be due to the change in resolution during the conversion between the Tif file and word/PDF. In the revised manuscript, we included a better Figure 2 with a higher resolution. In addition, we will upload high-resolution figures separately. We added a new supplemental figure (Fig. S2d) with higher resolution and magnification to show the presence of micro and macro lipid droplets and foamy macrophages in the lungs.
Comment: As ATGL contributes to the development of insulin resistance, we assume that ATGL and HOMA should follow the same direction. In Fig 1, HOMA (insulin resistance) is not related to MTB infection but related to HFD in juvenile mice. Why can we see the difference in ATGL after MTB infection in Fig S3a?
Response: We identified a mistake in presenting ATGL data in WAT in adult mice. In fact, ATGL significantly increased only in the infected HFD-fed mice compared to control mice (Figure 5), whereas another lipase (pHSL) significantly increased in infected RD-fed mice compared to control mice. Based on these data, in the revised manuscript, we corrected the Results section and revised the Discussion. Lipolysis and loss of lipid droplets contribute to the development of HOMA [PMID: 33097799]. Juvenile mice show no significant increase in lipases. However, there is a significant increase in p-perilipin 1, which indicates a loss of lipid droplets. It should be noted that the amount of body fat in adult mice is greater compared to juvenile mice and thus the acute loss of fat cells (increased degradation of lipid droplets) would be higher in adult mice compared to juveniles. Our data indicate that increased loss of lipid droplets/adipocytes may contribute to the risk of developing insulin resistance, similar to previous observations [PMID: 33097799].
Comment: In Fig S4, the responses of IFN-g and TNF-a to diet and MTB appear different between juvenile and adult mice. What’s the possible mechanism to explain the differences between juvenile and adult mice.
Response: Adult mice showed an elevated amount of body fat compared to juvenile mice. HFD significantly increased body fat in adult mice compared to juvenile mice. The changes in IFN-g and TNF-a in adipose tissue depend on the levels and activation of infiltrated immune cells during Mtb infection. Adult mice showed increased levels of lipases and more damage (histology) compared to juveniles during Mtb infection. It is known that diet alters the activation levels of immune cells such as CD8 and macrophages. Our data demonstrated increased TNF in WAT in RD-fed mice and not in HFD-fed mice during infection in adult mice. This suggests that a carbohydrate-rich diet activates macrophages and creates a pro-inflammatory local environment in adult mice.
Comment: In Fig 1, in uninfected adult mice, HFD is associated with an increase in HOMA. However, in Fig 5, there is no difference in ATGL between RD and HFD mice uninfected to MTB. What is the explanation?
Response: In uninfected HFD-fed mice, the development of insulin resistance does not depend only on the levels ATGL induced lipolysis (which increases lipid accumulation in other organs). However, in HFD-fed mice, increased circulating lipids (via diet) can elevate abnormal insulin levels, which could cause insulin resistance, leading to increased HOMA levels [PMCID: PMC5398492].
Comment: In Figure 7~10, the authors investigated the expression of markers related to lipid metabolism in lung tissue from adult mice. I would recommend authors provide the results in juveniles to confirm the role of aging in MTB-related lipid metabolism in lung tissue.
Response: We investigated the effect of loss of lipids in adipose tissue during Mtb infection on lung energy metabolism because our data indicated systemic insulin resistance in Mtb infected adult mice (and hence we have not performed western blot analysis in the lungs of juvenile mice). As we have discussed in the manuscript, an acute loss of lipid droplets elevates lipid levels in other organs including the lungs. In juvenile mice, we did not see a significant loss of body fat because the mice were much younger and have less body fat compared to adult mice. We provide sufficient data related to adipose tissue to compare juvenile and adult mice.
Comment: The expression of ATGL/P-HSL/P-perilipin in lung tissue and WAT in adult mice seems does not follow a similar pattern (Fig 5 and Fig 10). Can the authors explain the possible differences between lung tissue and WAT in lipid metabolism?
Response: Adipocytes and adipose tissue regulate whole-body energy homeostasis. They store excess energy in the form of triglycerides and release fatty acids via lipolysis for usage by other organs. Under physiological conditions, the expression levels of lipases in adipose tissue are tightly regulated. The lungs, where glucose serves as the primary source of energy, are physiologically distinct from adipose tissue: whereas adipocytes can store excess lipids, excessive accumulation of lipids in the lungs causes lipotoxicity. The lungs express ATGL/HSL and perilipin to catalyze the accumulated lipids.
Comment: To confirm the finding, I would recommend authors to confirm their findings in the levels of RNA expression in key markers related to lipid metabolisms, such as PPARa, ATGL, etc.
Response: As mRNA is eventually translated into protein, it is usually assumed that there is some sort of correlation between the levels of mRNA and protein. However, due to the various factors associated with post-translational modifications, it has been shown that the correlation between expression levels of protein and mRNA in mammals is relatively low. Moreover, the activation levels of many proteins involved in lipolysis (p-HSL and p-perilipin-1) and energy metabolism (p-AMPK, p-AKT, p-mToR) depend on their phosphorylation status. Therefore, we think the provided protein data is adequate (and better than RNA analysis) to support our studies.
Comment: In the final section, the authors speculated that weight loss in MTB individuals can be a risk factor for developing metabolic syndrome and diabetes. However, the correlation between weight loss and lipid metabolism and insulin resistance is not well investigated in mice and humans. I do not think this speculation is appropriate and should be removed.
Response: We appreciate the concerns of the reviewer. In the revised manuscript, we modified the text as follows: “Our data also suggest that an increased lipolysis/acute loss of adipocytes in adipose tissue and altered lipid metabolism in other organs including the lungs due to an increased accumulation of intracellular lipids may drive the pathogenesis of insulin resistance during Mtb infection”. This conclusion is based on many published articles [PMCID: PMC4038351; PMID: 21498783]. We also added the following sentences. In addition, the proinflammatory adipose tissue may contribute to systemic insulin resistance in RD- fed adult Mtb infected mice. Therefore, we conclude that deregulated adipocytes physiology is associated with the development of insulin resistance during Mtb infection. Our study warrants further investigation of the role of adipose tissue and loss/gain of fat mass in clinical subjects to understand the link between Mtb infection and metabolic syndrome.
Round 2
Reviewer 2 Report
All the questions raised by the reviewer are well answered. Congratulation to the authors for their great work.